Whole blood fatty acid concentrations in the San Cristóbal Galápagos tortoise (Chelonoidis chathamensis)

Dass Khushboo 1
Lewbart Gregory A. greg_lewbart@ncsu.edu 1
Muñoz-Pérez Juan Pablo 2 3
Yépez Maryuri I. 4
Loyola Andrea 4
Chen Emile 5
Páez-Rosas Diego 3
1 Department of Clinical Sciences, College of Veterinary Medicine, North Carolina State University , Raleigh , NC , United States of America
2 Faculty of Science and Engineering, University of the Sunshine Coast , Sippy Downs , Queensland , Australia
3 USFQ & UNC-Chapel Hill Galápagos Science Center (GSC), Universidad San Francisco de Quito , Quito , Ecuador
4 Direcion Parque Nacional Galápagos , Islas Galápagos , Ecuador
5 9 Oneida Court , Chester Springs , PA , United States of America
Schuster Richard
Electronic publication date: 2021 Jun 30
Publication date: 2021
Volume: 9
Electronic Location ID: e11582
Received 2020 Sep 14; Accepted 2021 May 19
Copyright: ©2021 Dass et al.
Copyright year: 2021
Copyright holder: Dass et al.
License: This is an open access article distributed under the terms of the Creative Commons Attribution License, which permits unrestricted use, distribution, reproduction and adaptation in any medium and for any purpose provided that it is properly attributed. For attribution, the original author(s), title, publication source (PeerJ) and either DOI or URL of the article must be cited.
License URL: https://creativecommons.org/licenses/by/4.0/

Keywords: Galápagos tortoise, Chelonoidis chathamensis, Fatty acid, Blood

Funding: The authors received no funding for this work.

==============================
To continue releasing San Cristóbal Galápagos tortoises housed in managed-care facilities at the Giant Tortoise Breeding Center of Galápagos National Park (Galapaguera de Cerro Colorado) to the Otoy Ecological Farm, health assessments and physical examinations were conducted. As a part of these wellness examinations, blood was drawn from 11 tortoises to analyze fatty acid concentrations. Fatty acid levels can provide insight into the nutritional profiles, immune status, and reproductive health of vertebrates. To the co-author’s knowledge, there is no current information about fatty acids in this species. It was hypothesized that there would be inherent differences based on the different geographic ranges, diets, sex, and age of turtles. It was noted that the ω-6/ω-3 ratio was higher for the breeding center than for the ecological farm and that overall polyunsaturated fatty acids (PUFAs) did not have any significant differences. The ω-6/ω-3 findings can contribute to a global picture of these fatty acids across taxa, as reptiles are underrepresented in this area of research. Additional results are a resourceful starting point for future investigations into how fatty acids are affected in Galápagos tortoises.

Introduction

The San Cristóbal Galápagos tortoise (SCGT), Chelonoidis chathamensis, is endemic to one of the oldest islands in the Galápagos Archipelago (Parent, Caccone & Petren, 2008; Miller et al., 2018). As the human population in the Galápagos Islands continues to expand, the SCGT faces growing threats and has been categorized as Endangered (EN) in the IUCN red list (Caccone et al., 2017).

Recent population genetic analyses have exposed that the population is recovering from a severe bottleneck that depleted its genetic diversity, with a population trend increasing of 2,950 adult individuals (Garrick et al., 2014). The species is protected for any commercial use and international trade under CITES Appendix 1 (Turtle Taxonomy Working Group, 2017). In an effort to restore their population, the SCGT has been brought into managed facilities for care, propagation, and reintroduction into the wild. Maintaining proper husbandry and conducting frequent health screenings at these facilities are essential components to help increase the chance of successful breeding and restocking programs.

Many health assessments and annual physical examinations include blood collection and analysis for standard hematology and biochemistry parameters (Geffré et al., 2009; Zhang, Gu & Li, 2011; Gibbons, Klaphake & Carpenter, 2013; Adamovicz et al., 2015; Lewbart et al., 2018; Carreta et al., 2019). Fatty acid (FA) profiles are not commonly included in health assessments of this or other tortoise species, so reference values have not been established. This study presents the fatty acid profile of C. chathamensis from both a captive breeding center and ecological farm where SCGT has been introduced for a conservation and ecotourism initiative and evaluates potential connections to nutritional status, immune function, and reproductive capabilities (Cartland-Shaw et al., 1998; Rustan & Drevon, 2005; Samee, Mantieghi & Estevez, 2019).

Materials & Methods

Sampling and study area

For the present study, 16 SCGT were sampled: eleven from the Otoy Ecological Farm (Finca Voluntad de Dios, Cerro Verde, Isla San Cristóbal, Islas Galápagos, Ecuador) and five from Giant Tortoise Breeding Center of Galapagos National Park (La Galapaguera de Cerro Colorado, Isla San Cristóbal, Islas Galápagos, Ecuador). All but one of the SCGT from the Otoy Ecological Farm were male. Alternatively, most of the SCGT from the Breeding Center were juveniles and thus undetermined sex, except for one that was male. All tortoises included in this work were clinically healthy as determined by physical examination, carapace and plastron measurements, blood collection (PCV/TS and biochemistry analysis), temperature, and fecal collection. Samples were collected in the springtime (March, 2018). Typical weather conditions during this time are 25−30 °C with sunshine and minimal cloud coverage. This study was performed as part of a population health assessment authorized by the Galapagos National Park Directorate (permit Nos. PC-21-18, PC-57-19) and approved by the Galápagos Science Center and North Carolina State University ethics and animal handling protocol. All handling and sampling procedures were consistent with standard vertebrate protocols and veterinary practices. No experiments were performed and such work is rarely if ever permitted in the Galápagos. Having access to these animals is a privilege and we were taking advantage of a valuable permit related to a health assessment to obtain as much information as possible form these animals while handling and taking blood samples. The data collected are simply a quantification of the fatty acids in this species of Galápagos tortoise.

Both sampling sites are located on the Northeast side of San Cristóbal Island and share some similarities. The “Galapaguera de Cerro Colorado” is one of a few sanctuaries that breeds Galápagos giant tortoises in captivity. It encompasses 11 hectares (30 acres) and holds 180 free-ranging SCGT that are either in the breeding program or are the offspring of these breeding pairs. The SCGT are allowed to roam freely and have access to a vast quantity of native vegetation, i.e., Manzanillo (Hippomane mancinella).

In comparison, the Otoy farm is situated on 16 hectares of private land where SCGT roam freely over an area of 1 and a half hectare with sparsely native plants and introduced trees. The SCGT are fed a mixture of otoy (Otoy yautia) and elephant grass (Pennisetum purpureum) at 5 kg/tortoise every other day. Additionally, there is a diverse group of non-native fruits at the site including passion fruit (Passiflora edulis), banana (Musa paradisciaca), guava (Psidium guaiva), plum (Spondias movie) and orange (Citrus aurantium) that the tortoises could have access. A similarity between the diets of both groups is that they have access to opuntia cactus (Opuntia echios).

Blood sample collection and handling

The SCGT chosen to be analyzed as a part of this study were part of an annual continuing health assessment performed as a part of pre-release screening. They were all healthy as determined by physical examination, fecal, and bloodwork parameters. Blood was collected from either the brachial sinus or dorsal coccygeal vein using a heparinized (Heparin Sodium USP, 1,000 units/ml; APP Pharmaceuticals, LLC, Schaumburg, IL, USA) 3.0 ml syringe with a 1.0″ 22-gauge needle. Two to three drops of whole blood were placed onto a paper card containing anti-oxidant stabilizing agents (PerkinElmer 226 Spot Saver RUO Card; PerkinElmer Health Sciences, Inc., Greenville, SC, USA) and allowed to air dry. Samples on the filter paper were stored at room temperature during transportation. Whole blood was collected and applied to a stabilized Perkin Elmer 206 blood spot card.  The card was allowed to dry and then mailed to Lipid Technologies (1600 19th Avenue SW, Austin MN 55912) for analysis (Marangoni, Colombo & Galli, 2004). Dried blood spots were then punched out into a 16 × 100 mm screw-top test and subjected to direct derivatization with acidic methanol. The resulting fatty acid methyl esters were extracted with hexane and separated by capillary column gas chromatography on a Shimadzu GC2010 gas chromatograph equipped with a 30 m FAMEWAX (Restek, State College, PA) column.  Fatty acid methyl esters were identified by flame ionization detection. Retention times were compared to a mixed fatty acid methyl ester standard from NuChek Prep (Elysian, MN). 

Statistical analysis

Differences in fatty acids levels between the ω-6/ ω-3 ratio and PUFA were analyzed. Sampling sites were determined using a Wilcoxon test to determine if the pairs were statistically the same or not based on current literature. The Holm-Bonferroni correction was applied to set an acceptable p-value cut off, given the high number of comparisons performed and decrease false-positive results. All statistical analysis was performed in R (version 3.4.3; The R Foundation for Statistical Computing).

Table 1 Fatty acid profile as percent (%).

Total fatty acid profile (Percent, %) of San Cristóbal Galápagos tortoises, Chelonoidis chathamensis.

Fatty Acid	Median (Breeding center)	Mean (Breeding center)	Median (Ecological farm)	Mean (Ecological farm)	
14:0	2.16	2.10 ± 0.31	1.72	1.86 ± 0.38	
14:1	1.04	1.15 ± 0.85	1.15	1.05 ± 1.00	
15:0	0.93	1.03 ± 0.57	0.59	0.80 ± 0.54	
16:0	17.35	16.96 ± 1.61	16.38	16.52 ± 1.03	
16:1 ω-7	8.18	7.66 ± 1.07	9.50	8.98 ± 1.84	
17:0	0.05	0.13 ± 0.17	0.04	0.07 ± 0.08	
17:1	0.24	0.26 ± 0.21	0.18	0.13 ± 0.11	
18:0	9.03	8.95 ± 1.31	8.36	8.72 ± 1.16	
18:1 ω-9	45.13	44.52 ± 7.68	45.52	44.96 ± 6.37	
18:2 ω-6	3.82	4.54 ± 2.42	3.37	4.00 ± 1.84	
18:3 ω-6	0.06	0.10 ± 0.12	0.06	0.08 ± 0.08	
18:3 ω-3	1.59	1.95 ± 1.05	1.38	1.45 ± 0.87	
20:0	0.06	0.08 ± 0.06	0.15	0.13 ± 0.08	
20:1 ω-9	0.00	0.01 ± 0.01	0.04	0.03 ± 0.03	
20:1 ω-7	0.53	0.58 ± 0.36	0.80	0.82 ± 0.30	
20:2 ω-6	0.05	0.06 ± 0.03	0.04	0.04 ± 0.01	
20:3 ω-9	0.23	0.23 ± 0.05	0.26	0.26 ± 0.10	
20:3 ω-6	0.51	0.55 ± 0.23	0.44	0.51 ± 0.16	
20:4 ω-6	2.81	3.23 ± 1.02	2.97	4.29 ± 2.35	
20:3 ω-3	0.04	0.04 ± 0.02	0.02	0.06 ± 0.08	
20:4 ω-3	0.27	0.23 ± 0.10	0.12	0.13 ± 0.12	
20:5 ω-3	2.33	2.12 ± 0.79	1.64	1.74 ± 0.61	
22:0	0.18	0.20 ± 0.11	0.24	0.21 ± 0.08	
22:1 ω-9	0.08	0.09 ± 0.02	0.11	0.08 ± 0.06	
22:4 ω-6	0.39	0.46 ± 0.17	0.54	0.57 ± 0.23	
22:5 ω-6	0.13	0.14 ± 0.06	0.08	0.09 ± 0.04	
22:5 ω-3	2.04	2.19 ± 0.82	1.89	1.85 ± 0.41	
24:0	0.04	0.04 ± 0.04	0.08	0.08 ± 0.07	
22:6 ω-3	0.13	0.16 ± 0.08	0.15	0.21 ± 0.18	
24:1	0.257	0.22 ± 0.13	0.26	0.26 ± 0.07	
Saturates	29.68	29.37 ± 3.16	28.56	28.32 ± 1.96	
Monoenes	47.43	46.57 ± 7.75	46.59	47.21 ± 5.86	
PUFA	15.05	16.01 ± 5.71	13.44	15.28 ± 4.87	
HUFA	9.49	9.36 ± 2.44	8.45	9.71 ± 3.33	
T/T	0.08	0.08 ± 0.03	0.06	0.07 ± 0.03	
Tot. w ω-3	7.82	6.69 ± 2.23	5.01	5.44 ± 1.57	
Tot. w ω-6	7.82	9.08 ± 3.76	7.72	9.58 ± 4.22	
Tot. ω-9	47.06	45.99 ± 7.44	45.91	46.38 ± 5.58	
ω-6/ ω-3	1.35	1.36 ± 0.35	1.90	1.84 ± 0.78	
AA/EPA	1.71	1.58 ± 0.26	2.80	2.59 ± 1.31	
ω-3 HUFA	49.86	50.45 ± 4.68	39.61	43.06 ± 9.69	
ω-6 HUFA	50.14	49.55 ± 4.68	60.39	59.94 ± 9.69	
EPA + DHA	2.44	2.28 ± 0.86	1.78	1.95 ± 0.65	

Results

The present data shows the median values for the full fatty acid profile from both environments, the Ecological Farm and Breeding Center (Table 1). The ω-6/ ω-3 ratio showed a significant difference (p-value = 0.014) with the group from the breeding center being higher than the ranch (2.023 vs 1.133, respectively) (Fig. 1). Comparison of PUFA between the center and the ranch (15.958 vs 13.954, respectively) was not significantly different (p-value = 0.563) (Fig. 2).

Figure 1 Fatty acid ω − 6∕ω − 3 ratio in Galapagos tortoise.

Fatty acid concentrations of ω − 6∕ω − 3 ratio in San Cristóbal Galapagos tortoise,Chelonoidis chathamensis.

Figure 2 Polyunsaturated fatty acids in Galapagos tortoises.

Polyunsaturated fatty acids concentrations in San Cristóbal Galapagos tortoise, Chelonoidis chathamensis.

Discussion

There is no information regarding normal circulating levels of fatty acids in Galapagos tortoises. Many factors influence the levels of fatty acids including diet, age, physiological status, and stress (Clauss, Grum & Hatt, 2007; Joseph, Ackman & Seaborn, 1985; Cartland-Shaw et al., 1998; Samee et al., 2019), which make it difficult for inter-species extrapolation. These results can provide answers for improving husbandry of animals kept in managed care facilities. Reptiles have unique fatty acids that are important to their biology. Current information in this area has shown a trend of higher ω-6/ ω-3 ratio in-managed care settings or higher PUFA in wild reptiles (Cartland-Shaw et al., 1998; Clauss, Grum & Hatt, 2007; Joseph, Ackman & Seaborn, 1985).

Increasing research in FA effects in human and companion pet nutrition has led to an awareness of some potential adverse effects with dysregulation of ω-6/ ω-3 ratio. Studies in human nutrition have shown how the shift in the ω-6/ω-3 ratio from 1/1 to 5∕1–50∕1 has negative impacts on the function of the neural and cardiovascular system and diseases including cancer and arthritis (Simopoulos & Cleland, 2003). One study of companion animals examined FA levels in commercially prepared diets for cats and dogs (Lenox & Bauer, 2013). An increased ω-6/ ω-3 ratio in cats showed altered platelet functions and improved glucose control via decreased serum insulin. While in dogs, a decreased ω-6/ ω-3 ratio showed delayed wound epithelialization of wounds, decreased number of neutrophils and leukotrienes B4, decreased CD4+ T lymphocytes, and lower delayed-type hypersensitivity response (Lenox & Bauer, 2013). Our study shows significantly higher values of ω-6/ ω-3 in the breeding center compared to farm. This could be due to higher levels of ω-6 or lower levels of ω-3 in the diet. While these trends and adverse effects of ω-6/ ω-3 have been observed in mammals, it is not guaranteed to translate to reptiles. The animals in our study were all healthy and were not likely in an inflammatory state and thus this was not a confounding factor for the ω-6/ ω-3.

Highly unsaturated fatty acids (HUFA) are those that contain multiple double bonds in their carbon chains and therefore are a specific type of PUFA. It has been established that plants and phytoplankton are among the only organisms that can synthesis these PUFAs, making them an essential dietary requirement for most animals, including reptiles (Cartland-Shaw et al., 1998; Rustan & Drevon, 2005). In the present study, there were no significant differences between the two groups for PUFAs, although the breeding center did numerically have higher concentrations than the farm. This finding is in contrast to a study that examines dietary and plasma concentrations of fatty acids between wild and managed tuatara (Sphenodon punctatus) and showed higher plasma concentrations of ω-3 PUFAs of the wild group than the managed group (Cartland-Shaw et al., 1998). It is interesting to hypothesis why the ω-3 HUFAs of the SCGTs at the breeding center were at such high levels and whether this impacted any of their membrane functionality in terms of fluid balance, immune system, or reproductive capability.

Since several tortoises in the present study came from a breeding center, it is essential to discuss the role of FA in reproduction. There have been several reproductive studies conducted to examine the role of fatty acid concentrations on fertility in vertebrates (Cartland-Shaw et al., 1998; Lin et al., 2016; Tran et al., 2017; Samee et al., 2019). In one study using zebrafish, it was concluded that the addition of extra virgin olive oil extracted from Koroneiki that was rich in oleic acid (18:1 ω-9) and linoleic acid (18:2 ω-6) improved the reproductive capability of both males and females in terms of sperm quality (motility, fertility, and hatching), frequency of follicular development, and accelerated hatching (Samee, Mantieghi & Estevez, 2019). Another study looked at the importance of ω-3 and ω-6 PUFA for male reproductive health in ruminants, including parameters of sperm membrane integrity, maintaining viable sperm during the chilling and freezing process, and testicular development (Tran et al., 2017). The ω-3/ ω-6 ratio in boar’s was examined and determined to improve testicular development and possibly chances for breeding selection overall libido in animals fed a 1:1 ω-3/ ω-6 ratio (Lin et al., 2016). While these references are promising developments to be correlated to reproductive success, the SCGT values in the present study were not examined for fecundity. This may be a future avenue to explore once normal FA have been established for this species.

Conclusions

A unique opportunity to analyze fatty acids vales for healthy Galapagos tortoises endemic to to San Cristóbal Island was incorporated into the annual health examinations of this population. This novel technique is the first time such a study has been performed in this species. This study supported this finding for the ω-6/ ω-3 ratio being higher in-managed care animals compared to wild animals. The results did not support the findings of overall PUFAs higher for in-managed care animals. Additionally, parameters influencing the differences of these fatty acids such as diet, sex, and physiologic status, could be examined.

Supplemental Information

Supplemental Information 1 June 2018 Fatty Acid Values

Raw data for tortoises sampled.

Click here for additional data file.

Supplemental Information 2 Tortoise fatty acid samples run October, 2018

This Excel file contains individual tortoise fatty acid values.

Click here for additional data file.

We wish to thank the following people for their support and assistance: Sofía Tacle, Carlos Mena, Stephen Walsh, Philip Page, Kent Passingham, Soledad Sarzosa, Ana Carrión, Sylvia Sotamba and Doug Bibus. Also, we thank the Galapagos National Park Directorate (GNPD) for the permits granted for sampling, and Galapagos Science Center (GSC) for the logistic support during the study. Finally, we thank the GNPD rangers of San Cristobal Island Technical Office, especially Rafael Díaz and Jeffrey Málaga, for their help and assistance in collecting data.

Additional Information and Declarations

Competing Interests

Author Contributions

Animal Ethics

Field Study Permissions

Data Availability

The authors declare there are no competing interests.

Khushboo Dass conceived and designed the experiments, analyzed the data, prepared figures and/or tables, authored or reviewed drafts of the paper, and approved the final draft.

Gregory A. Lewbart conceived and designed the experiments, performed the experiments, authored or reviewed drafts of the paper, and approved the final draft.

Juan Pablo Muñoz-Pérez performed the experiments, authored or reviewed drafts of the paper, and approved the final draft.

Maryuri I. Yépez and Andrea Loyola conceived and designed the experiments, authored or reviewed drafts of the paper, and approved the final draft.

Emile Chen analyzed the data, prepared figures and/or tables, authored or reviewed drafts of the paper, and approved the final draft.

Diego Páez-Rosas conceived and designed the experiments, analyzed the data, authored or reviewed drafts of the paper, and approved the final draft.

The following information was supplied relating to ethical approvals (i.e., approving body and any reference numbers):

The Galapagos Science Center (GSC) does not ask for an ethics and wildlife management protocol. However, we follow and respect the stringent regulations and protocols that the Directorate of the Galápagos National Park (DPNG) and the Ministry of the Environment of Ecuador (MAE) have in place to be able to conduct research projects in the archipelago. Those protocols are stated at each research permit and are reviewed every year by the DPNG.

The research has been carried out as part of a research project authorized by the Directorate of the Galápagos National Park (DPNG) under the permit PC-57-19 “Evaluación del estado de salud de las tortugas gigantes de la Galapaguera de Cerro Colorado, previo a los procesos de liberación y repatriación” granted to Dr.Gregory Lewbart, Diego Páez-Rosas, PhD and Juan Pablo Muñoz-Pérez, PhD (S) through the Galápagos Science Center (GSC) respecting all the DPNG regulations.

In this context, the GSC certifies that all proposed techniques, samples, handling and, sampling procedures have been and will be consistent with standard vertebrate protocols and safe and appropriate practices for wild animals.

The following information was supplied relating to field study approvals (i.e., approving body and any reference numbers):

The Galapagos National Park Directorate approved the field study (No. PC-57-19).

The following information was supplied regarding data availability:

All the raw data with the fatty acid concentrations from each tortoise are available in the Supplemental Files.

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
