# Peer review of "Whole blood fatty acid concentrations in the San Cristóbal Galápagos tortoise (Chelonoidis chathamensis)"

_PeerJ, doi:10.7717/peerj.11582_

## Round 0.1 · original submission · Major Revisions

We have received two in-depth reviews of your paper, indicating that the paper is an interesting contribution, but that there several points that have been raised need to be addressed in a revision.

I would also like to note that the tone of reviewer 1 is not very collegial in places, for which I would like to apologize. Please try to look past the sometimes confrontational tone and try to focus on their substantive comments when you prepare your revision.

Reviewer 1 ·

Basic reporting

1) I can say nothing about the linguistic shortcomings because several of co-authors are from universities in USA or Australia.

2) I could not find the hypothesis behind the work in the section "Introduction". Prior literature has not appropriately been used to prepare the section Introduction and to discuss findings.

3) I think the paper structure follow the journal guidance. Table should be improved, in the General comments for the author I explained about that. Figures are redundant, they are presenting nothing more than the table, just a data repetition. Raw data are not bad.

Experimental design

1) I could not find a question in the paper. The paper tried to uncover a gap, but was not successful. When there is not a question without any hesitation there is no an answer.

2) The work cannot address the minimum standards of a research program. In fact, this is not a research program, it is just a GC.

3) There are serious missing in the section Materials and Methods I explained some of them in the General comments for the author. The main technique of used in the work is GC although there is no description about that!

Validity of the findings

The simple GC of carried out in the work cannot support the matters claimed in the discussion and conclusion. Generally, the sections are prepared based on the section "result" and the early studies, but I didnt find any result in the work, I saw just a GC output .

Additional comments

This is a work in which authors collected blood from several tortoises (without any specimen selection criteria) and then the blood samples subjected to a GC to determine their fatty acid composition.

General idea

 The surprising point is that the work have seven co-authors. What is their roles in the GC?

 I provided the work with a series of comments although the comments cannot improve the scientific level of the work.


Comments

Title
 This is not a title of a research project. This is just a title for a test (GC), nothing more.

Abstract
1) The abstract consist of 144 words. Seventy two words (50%, lines 27-33) of the 144 are background.

2) The abstract says nothing about the hypothesis behind the work.

3) Lines 33-34: “very little information about baseline values for fatty acids in this species exists”. → What does mean the phrase “very little”? → This means there is few studies, if yes, why didn’t cite in the section discussion and didn’t consider them to discuss your findings.

4) Lines 34-36: “It was noted that the ω-6/ω-3 ratio was higher for the breeding center than for the ecological farm and that overall polyunsaturated fatty acids (PUFAs) did not have any significant differences”. → This doesn’t show the result of the work, this is just a GC output.

5) Lines 36 – 38: “This study presents these fatty acid profiles to establish this information and allow future studies to continue expanding the understanding of their influence on husbandry in managed-care facilities”. → This is an irrelevant and redundant sentence for abstract. This can be used in the end of the section introduction. In the part of the abstract, authors should add a sentence that in which the general concept of authors about their results is presented.

Introduction
The end of the section is important for me where authors should clearly let readers know the hypothesis behind the work, but I could not find their hypothesis for the work. They just say that “this study presents the fatty acid profile of C. chathamensis from both a captive breeding center and ecological farm”. In fact, they say that they just determined the blood FA composition of several specimens.




Materials and Methods
1) Lines 70 – 95: Please provide the section with a map and indicate the sampling sites on it.

2) Lines 70 – 95: When the samples were collected? (sampling time - season)

3) Lines 70 – 95: Please add the environmental parameters in the sampling sites.

4) Line 70 – 95: Please provide the section with the criteria that have been considered to select the specimens.

5) Line 70 – 95: Please let us know the age, size, and the sex (I think all were male) of the specimens of selected.

6) Lines 104 – 107: This is not enough, authors should provide the section with details about lipid extraction, methylated esters preparation, and GC, and the methods should be provided with suitable references.

Results
1) It is better to delete the columns related to median of the FA values in the table 1.

2) What is the unit of the FA values in the table 1?

3) Please add the FA values as mean±SD

4) In the footnote of the table 1 you need to add that results sharing the same superscript letter are not statistically significantly different.

5) The FA values as biochemical data should be correlated to various endpoints at molecular and organismic levels to uncover their effects on the biology of the animals, otherwise the values are just accounted for a summarize of the GC out put.

Discussion
1) Lines 127-149: the lines are just a story, irrelevant and redundant.

2) Lines 149-150: a repetition from the result section.

3) Line 151: the data presented in the current paper cannot support the claim. To support the claim the diet FA profile should be determine.

4) Lines 151-153: Since most models that have studied adverse effects of ω-6/ω-3 are mammalian, extrapolations to reptiles must be drawn cautiously. → What does mean the sentence?

5) Lines 153-154: “it is likely that this altered ω-6/ω-3 would have some effect on immune system regulation and the inflammation cascade in the SCGT from the present study”. → You cannot approve the claim just based on a rough GC output.

6) Lines 155-156: “the animals in our study were all healthy and were not likely in an inflammatory state”. → What were the health criteria? Please introduce them in the “materials and methods” and then present related data in the “result” section.

7) Lines 156-157: “if these animals were from a rehabilitation facility, there could be implications for alterations to this ω-6/ω-3 ratio”. → What does mean the sentence?

8) Lines 159-162: “highly unsaturated fatty acids (HUFA) are those that contain multiple double bonds in their carbon chains and therefore are a specific type of PUFA. It has been established that plants and phytoplankton are among the only organisms that can synthesis these PUFAs”. → This is an old story, irrelevant and redundant.

9) Lines 169-172: “it is interesting to hypothesis why the ω -3 HUFAs of the SCGTs at the breeding center were at such high levels and whether this impacted any of their membrane functionality in terms of fluid balance, immune system, or reproductive capability. → Your data cannot support the claim.

10) Lines 174-191: “since several tortoises in the present study came from a breeding center, it is essential to discuss the role of FA in reproduction. There have been several reproductive studies conducted to examine the role of fatty acid concentrations on fertility in vertebrates (Cartland-Shaw et al., 1998; Lin et al., 2016; Len Van Tran et al., 2017; Samee et al., 2018). In one study using zebrafish, it was concluded that the addition of extra virgin olive oil extracted from Koroneiki that was rich in oleic acid (18:1 ω-9) and linoleic acid (18:2 ω-6) improved the reproductive capability of both males and females in terms of sperm quality (motility, fertility, and hatching), frequency of follicular development, and accelerated hatching (Samee et al. 2018). Another study looked at the importance of ω-3 and ω-6 PUFA for male reproductive health in ruminants, including parameters of sperm membrane integrity, maintaining viable sperm during the chilling and freezing process, and testicular development (Len Van Tran et al., 2017). The ω-3/ω-6 ratio in boar’s was examined and determined to improve testicular development and possibly chances for breeding selection overall libido in animals fed a 1:1 ω-3/ω-6 ratio (Yan Lin et al., 2016). While these references are promising developments to be correlated to reproductive success, the SCGT values in the present study were not examined for fecundity. This may be a future avenue to explore once normal FA have been established for this species”. → Irrelevant and redundant, this is just a story.

Conclusion
1) Lines 194-195: “the objective of the study was to present FA values for the Galapagos Tortoise endemic to San Cristóbal Island”. → This is not an objective.

2) Lines 195-197: “previous studies conducted in other reptilian species showed patterns of higher overall PUFAs and ω-6/ω-3in-managed animals compared to wild animals. → In the conclusion section you should summarize your findings no discuss them”. Please delete the sentence.

3) Lines 199-200: “future studies could examine a larger population to validate baseline FA values further. Additionally, parameters influencing the differences of these fatty acids such as diet, sex, and physiologic status, could be examined”. → In a weak work you cannot talk about outlooks

Reviewer 2 ·

Basic reporting

No comments

Experimental design

The research question (use of fatty acid profile to check the nutritional status, immune function and reproductive capabilities of turtles) is well formulated
The experimental design and methods used for taking the samples and for lipid and fatty acids analyses are correct, as well as the statistical analysis. Ethical standards are defined. Method for fatty acid analysis is just cited (AOCS, 2005) and carried out in an external laboratory

Validity of the findings

The findings are too basic to get any new information about the nutritional and immune status of the turtles, neither to assess the reproductive capability. There is no much information published about the fatty acid profile in turtles but the resutls obtained are not easy to discuss, it is just a list of fatty acids found in the blood without knowing which diet the turtles have taken (ingested) or what was the reproductive status (immature, mature) or sex (females, males). The authors centered most of the discussion in the ration n-6/n-3 but in some animals (i.e. fish) a high ratio means low n-3 and high n-6 and that is not good at all for the animals. For mammals is different but we are talking about aquatic animals, reptles.
In marine animals N-3 (omega 3) fatty acids are very important for neural development, growth and health status. In marine turtles should be more or less the same, depending on their food ingested (micro and macro algae have a lot of n-3 fatty acids). In this case the higher proportion of fatty acids came from n-6, and mostly arachidocnic acid (involved in reproduction in marine fish) and linolenic acid (18:2n-6) typical from plants.
As a summary the results show the first fatty acid profile of these turtles as a first step but it should have included a comparison not only between stocks (farmed and kept in a breeding center) but also between different diets used for feeding and between different reproductive status or size of the animals. It is too basic....

Additional comments

Continue with the work, this is just a preliminary analysis of what can be done, but the information provided is too basic and without any conclusive results to be used neither for growth of the animals , for reproduction or to assess the health status. It is just showing the fatty acid profile in the blood.

---

## Round 0.2 · Minor Revisions

I have had a look at the rebuttal letter and noticed that responses to reviewer comments were missing in several places.

I also found the coloring choice and the number of colors (>4) confusing.

Could you please address all reviewer comments and make sure that if you do use colors to distinguish reviewer comments from your responses stick to two colors.

At this point the rebuttal letter does not seem finished and I would ask that you clean it up and finish it before we can send your manuscript out for review again.

Thanks,
Richard

---

## Round 0.3 · Minor Revisions

Thank you very much for submitting your revised manuscript and supporting documents.

We've had a really hard time finding reviewers for this revision, which is why we eventually decided that I would review your revisions myself.

Thank you for your point by point responses to reviewer comments from your initial revision.

You have addressed all comments to my satisfaction. There is only one small change that I would ask of you.

You have responded to the lack of clarity in the methods with a detailed description in the rebuttal letter that does not seem to have made it into the manuscript. Could you please add this description to the manuscript or add this as supplementary material? I think this would be useful information for readers to see. Thanks!

---

## Round 0.4 · Minor Revisions

Thank you for submitting a revised version of your manuscript.

Could you please address the comments made by one of our section editors on your latest submission:

"In this new version, the methods for how the fatty acids were measured is nowhere in the text, and instead the authors have responded to this latest request by simply adding 3 sentences: "No experiments were performed and such work is rarely if ever permitted in the Galápagos. Having access to these animals is a privilege and we were taking advantage of a valuable permit related to a health assessment to obtain as much information as possible form these animals while handling and taking blood samples. The data collected are simply a quantification of the fatty acids in this species of Galápagos tortoise."

It does not matter whether there were experiments or not - data were collected and the methods should detail how exactly those data were collected so that someone else could replicate the results. The referees have objected to a lack of detail on how the fatty acids were quantified - if it is a simple quantification of the fatty acids (as the authors respond above), then the authors need to tell us EXACTLY how that was done.

The methods for the paper contain almost 2 pages on the sampling sites and study area (lines 72-108), another section on the handling of the animals and collection of blood (lines 111-123) and a short section on statistical analyses. In contrast, there is no section on how fatty acids were quantified at all, and the only reference I can find as to how the data reported in the paper were collected is given a total of 16 words at the end of the blood section: "the whole blood spot cards to be prepared into methylated esters and analyzed using gas chromatography (AOCS, 2005)."

It is impossible to see how anyone could replicate this work from the information provided and as such it does not meet the standards of the journal in my opinion. PeerJ policies are pretty clear - the data must be made available and the methods and data analyses must be described in sufficient detail that the study could be replicated if someone wished to do so. If the authors cannot provide detailed methods to document how the data presented in the manuscript were collected (other than "by gas chromatography"), then the data do not meet the standards for publication as submitted."

---

## Round 0.5 · accepted · Accept

Thank you for adding the details on the methods.
In my view, your manuscript is now ready for publication.